# Examining Association of Personality Characteristics and Neuropsychiatric Symptoms in Post-COVID Syndrome

**DOI:** 10.3390/brainsci12020265

**Published:** 2022-02-14

**Authors:** Cristina Delgado-Alonso, María Valles-Salgado, Alfonso Delgado-Álvarez, Natividad Gómez-Ruiz, Miguel Yus, Carmen Polidura, Carlos Pérez-Izquierdo, Alberto Marcos, María José Gil, Jorge Matías-Guiu, Jordi A. Matias-Guiu

**Affiliations:** 1Department of Neurology, Hospital Clínico San Carlos, Health Research Institute “San Carlos” (IdISSC), Universidad Complutense de Madrid, 28040 Madrid, Spain; cristinadelgado1409@gmail.com (C.D.-A.); mariavsgm@hotmail.com (M.V.-S.); alfonso.delgado.alvarez@hotmail.com (A.D.-Á.); amarcosdolado@gmail.com (A.M.); mariajosemedcu@hotmail.com (M.J.G.); matiasguiu@gmail.com (J.M.-G.); 2Department of Radiology, Clínico San Carlos, Health Research Institute “San Carlos” (IdISSC), Universidad Complutense de Madrid, 28040 Madrid, Spain; vidiado@yahoo.es (N.G.-R.); miguel_yus@yahoo.com (M.Y.); mariacarmen.polidura@salud.madrid.org (C.P.); 3Department of Agricultural and Forestry Engineering, University Center of Plasencia, University of Extremadura, 10600 Plasencia, Spain; carlospi@unex.es

**Keywords:** COVID-19, post-COVID syndrome, personality, depression, cognitive

## Abstract

Background: We aimed to evaluate personality traits in patients with post-COVID syndrome, as well as the association with neuropsychiatric symptoms present in this disorder. Methods: The Big Five Structure Inventory was administered to 93 consecutive patients with a diagnosis of post-COVID syndrome as defined by the WHO and to demographically matched controls. We also performed a comprehensive evaluation of depression, anxiety, fatigue, sleep quality, cognitive function, and olfactory function. Results: Patients with post-COVID syndrome scored lower for emotional stability, equanimity, positive mood, and self-control. Extraversion, emotional stability, and openness correlated negatively with anxiety and depression levels. Conscientiousness correlated negatively with anxiety. No statistically significant correlations were observed between personality traits and cognitive function, sleep quality, olfactory function, or fatigue. Personality scores explained 36.3% and 41% of the variance in scores on the anxiety and depression scales, respectively. Two personality profiles with lower levels of emotional stability were associated with depression and anxiety. Conclusions: Our study shows higher levels of neuroticism in patients with post-COVID syndrome. Personality traits were predictive of the presence of depression and anxiety, but not cognitive function, sleep quality, or fatigue, in the context of post-COVID syndrome. These findings may have implications for the detection of patients at risk of depression and anxiety in post-COVID syndrome, and for the development of preventive and therapeutic interventions.

## 1. Introduction

The WHO has recently defined post-COVID-19 condition or post-COVID syndrome as a disorder occurring in patients with a history of n-SARS-CoV-2 (novel severe acute respiratory syndrome coronavirus 2) infection who present symptoms that cannot be explained by an alternative diagnosis [1]. These symptoms usually present 3 months after the onset of COVID-19 and last at least 2 months. According to the WHO, the most common symptoms include fatigue, shortness of breath, and cognitive dysfunction. Other common symptoms include depression, anxiety, headache, joint and muscle pain, sleep problems, and smell or taste disorders [2,3].

The pathogenesis of post-COVID syndrome remains unknown. Several mechanisms have been suggested, including prolonged inflammation, vascular injury and endothelial dysfunction, sequelae of organ damage, and the effects of hospitalisation [4,5,6]. Due to the heterogeneity of clinical symptoms, multiple mechanisms may be involved. Other explanations, such as viral induction of a neurodegenerative process, are under investigation [7,8].

Personality traits are stable characteristics that reveal patterns of behaviour, values, habits, feelings, and thoughts. Several models have been developed to describe and assess personality. Among them, the most widely used is the Big Five Structure Inventory (BFSI), which categorises the main personality traits into openness, conscientiousness, extraversion, agreeableness, and neuroticism [9].

Some studies have examined the relationship between personality traits and engagement with containment measures during the COVID-19 pandemic [10,11], the risk of depression and anxiety [12,13], and the psychological impact of the pandemic [14].

Furthermore, previous research has linked personality factors with certain diseases and clinical characteristics. Specifically, neuroticism, positive emotion in extraversion, and competence and self-discipline in conscientiousness were associated with anxiety and depression [9]. Patients with different forms of pain may exhibit certain personality traits or be more prone to chronic pain or developing reactive depression [15,16,17]. In addition, personality traits have been linked to subjective cognitive perception [18,19]. The role of personality has also been examined in other infections with potential neuropsychiatric involvement. For instance, openness was associated with better global cognitive function in HIV [20]. Furthermore, in patients living with HIV, higher neuroticism and lower extraversion were associated with poorer quality of life, and lower conscientiousness was linked to depression [21,22,23]. In patients with hepatitis C virus taking pegylated interferon and ribavirin, low self-directedness was associated with induced depression by the antiviral drugs [24]. However, to our knowledge, no study has evaluated the role of personality traits in post-COVID syndrome. The study of personality in the context of patients with post-COVID syndrome may be of interest for developing therapeutic strategies in case of maladaptive coping mechanisms. Investigating personality traits may also improve our understanding of the mechanisms of neurological and psychiatric symptoms in post-COVID syndrome. Personality traits may also promote behaviours (e.g., physical exercise, cognitive activities, and social isolation) that could positively or negatively influence the wide spectrum of post-COVID symptoms.

In this study, we evaluated the role of personality traits in patients with post-COVID syndrome. Firstly, we aimed to compare the personality traits of a cohort of patients with post-COVID syndrome and a group of controls, according to the BFSI. Secondly, we evaluated the correlation between the main personality traits and the neuropsychiatric features of post-COVID syndrome. Thirdly, we examined the association between personality profiles and the clinical characteristics of post-COVID syndrome.

## 2. Materials and Methods

### 2.1. Participants and Procedure

This study included 93 consecutive patients with post-COVID syndrome attended at our centre’s neurology department due to cognitive issues. Patients met the current criteria for post-COVID syndrome proposed by the WHO (World Health Organization, 2021). Patients with other diagnosis previous to the onset of COVID-19 and potentially associated with symptoms (e.g., neurological or psychiatric disorders) were excluded. The mean age was 50.39 ± 11.26 years; 66 patients (71%) were women. Mean time from COVID-19 onset to assessment was 11.20 ± 4.29 months. The main clinical characteristics are presented in Table 1. A group of healthy controls with no history of COVID-19 was also included. Patients and controls were matched 1:1 for age (<6 years) and sex. The mean age of the healthy controls was 50.28 ± 13.50; 66 participants (71%) were women; and mean years of schooling were 15.32 ± 3.30.

Personality was assessed using the BFSI, a multi-dimensional questionnaire based on five personality factors: emotional stability (inverse scores of neuroticism), extraversion, openness, conscientiousness, and agreeableness. Each factor is calculated from the parameters of six subscales (for instance, emotional stability is calculated from the carefreeness, equanimity, positive mood, social confidence, self-control, and emotional robustness subscales). For each item, the participant was asked to rate the accuracy of a statement using a four-point scale. The questionnaire was self-administered using the standard form of the test included in the Vienna Test System^®^ (Schuhfried GmbH; Mödling, Austria), and we ensured that all participants received the same information about the test, with no external influences. Raw scores were converted to percentiles, taking into account sex, education, and age.

Depression was assessed using the Beck Depression Inventory-II (BDI-II) [25], and anxiety using the State-Trait Anxiety Inventory (STAI) [26]. Sleep quality was examined with the Pittsburgh Sleep Quality Inventory. The Modified Fatigue Impact Scale was administered to assess fatigue [27]. BDI-II ≥ 19 was regarded as moderate or severe depression, and STAI-S ≥ 40 was considered clinically significant anxiety [25,26]. According to these cut-off points, 22% of patients were regarded as having moderate-severe depression, and 46% clinically significant anxiety. The Brief Smell Identification test was used to assess olfactory function. Patients also underwent cognitive assessment with a comprehensive neuropsychological protocol (Appendix A), including the following tests: forward and backward digit span, Corsi block-tapping test, Symbol Digit Modalities test, Boston Naming Test, Judgment of Line Orientation, Rey-Osterrieth Complex Figure (copy and recall at 3 and 30 min), Free and Cued Selective Reminding Test, verbal fluencies (animals, words beginning with “*p*”, “m”, and “r”), Stroop Color and Word Interference test, Visual Object and Space Perception battery (object decision, progressive silhouettes, number location, and position discrimination), Trail Making Test, Figural Memory Test, Tower of London, N-back verbal test, Cognitrone, Reaction Test, Determination Test, and WAF battery. This protocol has been described elsewhere [28].

All assessments were performed in person by a trained neuropsychologist.

### 2.2. Statistical Analysis

Statistical analysis was performed using SPSS Statistics 24.0 and R package version 3.6.3. Descriptive data are shown as mean ± standard deviation or median (interquartile range). The chi-square test was used to compare categorical variables. The two-sample t-test and ANOVA with Tukey post-hoc test were used to examine intergroup differences in continuous variables. *p* values < 0.05 were considered statistically significant. To compare personality factors and subfactors (35 variables), we applied a false discovery rate correction for multiple comparisons [29].

The two-tailed Pearson coefficient was used to evaluate correlations between quantitative variables in the post-COVID syndrome group. Correlations were regarded as weak (<0.30), moderate (0.30–0.49), or strong (>0.49), according to the correlation coefficient. Statistical significance was set at *p* < 0.01 to reduce the risk of multiple comparisons.

Automatic linear modelling (LINEAR) was performed to identify the personality traits that predict depression and anxiety. All factors and subfactors of BFSI were introduced in the model as predictors, and BDI-II and STAI state anxiety (STAI-S) scores were regarded as the independent variables. Only variables with *p* values < 0.05 were retained as predictors.

We used Ward’s linkage algorithm [30], an unsupervised method of agglomerative hierarchical clustering, to identify subtypes of patients according to the five main personality traits. This analysis was performed using data from both patients and controls.

## 3. Results

### 3.1. Comparison between Patients with Post-COVID Syndrome and Controls

Patients with post-COVID syndrome scored lower for emotional stability (Table 2). When examining all subfactors of the BFSI, patients with post-COVID syndrome presented lower scores for equanimity, positive mood, and self-control (Table 3).

### 3.2. Correlation Analysis

Conscientiousness showed negative correlations with STAI-S (r = −0.364, *p* < 0.001) and STAI trait anxiety (STAI-T) scores (r = −0.347, *p* = 0.001). Extraversion showed negative correlations with BDI-II (r = −0.326, *p* = 0.002), STAI-S (r = −0.374, *p* < 0.001), and STAI-T scores (r = −0.495, *p* < 0.001). Emotional stability was also negatively correlated with BDI-II (r = −0.360, *p* < 0.001), STAI-S (r = −0.342, *p* = 0.001), and STAI-T scores (r = −0.612, *p* < 0.001). Openness also showed negative correlations with BDI-II (r = −0.314, *p* = 0.002), STAI-S (r = −0.283, *p* = 0.006), and STAI-T scores (r = −0.297, *p* = 0.004). Agreeableness did not show a significant correlation with scores on any neuropsychological instrument.

No statistically significant correlations were identified between the five personality factors and fatigue, sleep quality, olfactory function, or objective cognitive testing. Neither did we observe any correlation with months from symptom onset to consultation. All correlations with BFSI factors and subfactors are shown in Figure 1, Figure 2 and Figure 3 and Appendix A.

### 3.3. Personality-Related Predictors of Depression and Anxiety

The results of automatic linear modelling are shown in Table 4. Regarding depression (BDI-II), linear modelling identified openness to ideas, obligingness, dynamism, and openness to feelings as significant predictors, and the model explained 36.3% of variance. For anxiety (STAI-S), the model identified openness to actions, caution, love of order, competence, cheerfulness, social confidence, adventurousness, discipline, assertiveness, and openness to aesthetics as predictors, and explained 41% of variance.

### 3.4. Cluster Analysis

The optimal cluster analysis solution was found at four clusters (Figure 4). The mean values of personality traits for each group are shown in Figure 5. Cluster one showed higher levels of depressive symptoms than clusters two and three. Regarding anxiety, STAI-S scores were higher in clusters one and four than in cluster two. No statistically significant differences were observed in fatigue or sleep quality (Table 5).

## 4. Discussion

In this study, we used the BFSI to evaluate personality traits in patients with post-COVID syndrome. We aimed to disentangle the personality characteristics of these patients and to clarify the association between certain personality traits and the neuropsychiatric symptoms of post-COVID syndrome. To our knowledge, studies have evaluated the role of personality traits in post-COVID syndrome.

Patients with post-COVID syndrome showed lower levels of emotional stability (higher neuroticism). In addition, the analysis of subfactors revealed lower scores for equanimity, positive mood, and self-control, all of which belong to the neuroticism/emotional stability factor. Accordingly, this trait would suggest greater tendencies to stress, worries, or anxiety. In this regard, the neuroticism factor showed strong and moderate correlations with STAI-T and STAI-S scores, respectively. Levels of conscientiousness and extraversion showed moderate correlations with anxiety and depression. Regarding the automatic linear analysis, the models identified several subfactors (e.g., openness to ideas, feelings, and actions; love of order; competence; etc) that have previously been associated with affective disorders [9]. Overall, these findings suggest that personality traits at least partially explain the presence of depressive and anxiety symptoms in the context of post-COVID-19 syndrome.

Interestingly, no statistically significant correlations were observed between personality traits and cognition, sleep quality, olfactory function, or fatigue. Similarly, there were no differences in these symptoms when comparing the personality profiles. This finding is noteworthy because it suggests that these symptoms are independent of personality traits. Cognitive and olfactory symptoms are often reported among the most frequent symptoms after acute infection [31]. The main cognitive domains involved in post-COVID syndrome are attention/executive functioning and episodic memory [28,32]. In this study, we included a comprehensive protocol for neuropsychological assessment including the main cognitive domains. In addition, olfactory function was quantified with a validated test. These methods should avoid the limitations of using brief cognitive assessments or self-reported cognitive or olfactory assessments. Although the association between personality traits and olfactory or cognitive function was not our hypothesis, we included them in the analysis to cover the wide range of neuropsychiatric symptoms in post-COVID syndrome. Furthermore, studies in other settings (e.g., aging and neurodegenerative disorders) have suggested a potential link between personality traits and cognitive performance [33,34,35,36], warranting examination in patients with post-COVID syndrome. The finding of statistically significant correlations with anxiety and depression, but not cognitive or olfactory function, suggest potential differences in the pathophysiology of the different symptoms present in post-COVID syndrome.

The analysis of the distribution of patients in four clusters suggests the following profiles: the first group, with lower scores in the main personality traits, especially extraversion, emotional stability, and openness; the second group, which may be identified as the resilient type according to the ARC typology [37]; the third group, with average levels but lower openness and higher emotional stability, which may be classified as reserved; and the fourth group, with lower emotional stability, which could be identified as the overcontrolled group. Groups one and four, both with lower levels of emotional stability, presented higher scores in BDI-II and STAI, confirming the vulnerability of these personality profiles to depression and anxiety in the context of the post-COVID syndrome.

Our study presents some limitations. Firstly, our controls had no history of COVID-19. A control group including patients affected by COVID-19 but without post-COVID syndrome would be of interest. Secondly, the personality assessment was performed at the time of assessment. Although evidence shows that personality traits are quite stable over time [38], we cannot exclude the possibility that diagnosis of COVID-19, the impact of the pandemic, or individual circumstances may induce changes in the response to the personality questionnaire. Thirdly, we did not consider in the analysis the severity of the disease, treatments used, hospitalization, time since diagnosis of COVID-19, and number of n-SARS-CoV-2 infections. Fourth, we could not add scores for the control group because the scales BDI-II, STAI, MFIS, PSQI, and BSIT were not administered to the control group. We could not conduct correlation analysis of selected variables on the control group. Thus, we could not ascertain between group statistical differences between the control and analysed group, which could bias our findings and interpretation. Fifth, we focused our analysis on the total score of the BDI-II. Future studies examining specific depressive symptoms may be of interest to examine the interplay between personality traits, mood, and depressive symptoms.

## 5. Conclusions

In conclusion, our study shows higher levels of neuroticism in patients with post-COVID syndrome compared to people who were never diagnosed with COVID-19. Several personality traits were predictive of the presence of depressive symptoms and anxiety. This supports the role of personality traits in coping behaviours during chronic disorders. Conversely, cognitive function, sleep quality, olfactory function, and fatigue were not associated with personality characteristics. These findings may have implications for the detection of patients at risk of depression and anxiety in the context of post-COVID syndrome, and for the development of preventive and therapeutic interventions.

## Figures and Tables

**Figure 1 brainsci-12-00265-f001:**
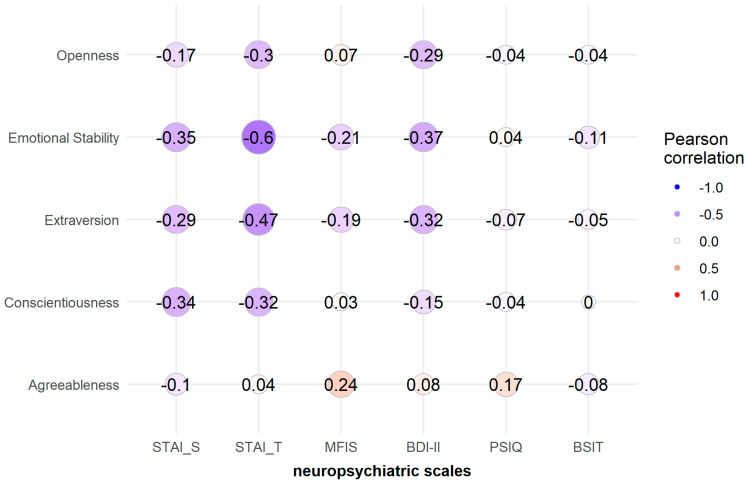
Heatmap of Pearson correlation coefficients between personality factors and scales.

**Figure 2 brainsci-12-00265-f002:**
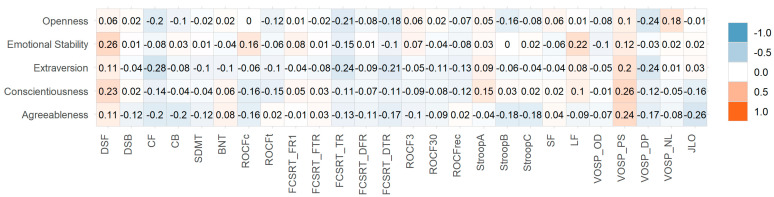
Heatmap of Pearson correlation coefficients between neuropsychological tests and personality factors. DSF: Digit span forward; DSB: Digit span backward; CF: Corsi forward; CB: Corsi backward; SDMT: Symbol Digit Modalities Test; BNT: Boston Naming Test; ROCF: Rey-Osterrieth Complx Figure (c: copy accuracy; t: time); FCSRT: Free and Cued Selective Reminding Test (FR1: Free Recall trial 1; FTR: Free Total Recall; TR: Total Recall; DFR: Delayed Free Recall; DTR: Delayed Total Recall); ROCF3: Rey-Osterrieth Complex Figure memory at 3 min; ROCF30: Rey-Osterrieth Complex Figure memory at 30 min; Stroop: Stroop Color Word Interference test (part A: word reading; part B: color naming; part C: interference); SF: Semantic fluency; LF (Letter fluency (*p*-words); VOSP: Visual Object and Space Perception Battery (OD: Object decision; PS: Progressive silhouettes; DP: discrimination of position; NL: number location); JLO: Judgment Line Orientation.

**Figure 3 brainsci-12-00265-f003:**
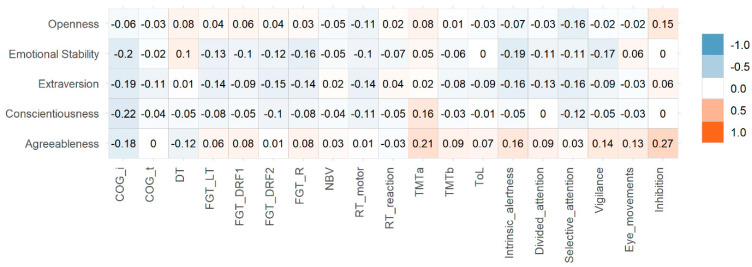
Heatmap of Pearson correlation coefficients between computerized neuropsychological tests and personality factors. COG: Cognitrone (i: incorrect reactions; t: mean time); DT: determination test; FGT: Figural memory test (LT: Learning total; DRF1: delayed free recall 1; DRF2: Delayed Free Recall 2; R: Recognition; NBV: N-back verbal test; RT: reaction test; TMT: Trail Making Test; ToL: Tower of London; intrinsic alertness, divided attention, selective attention, vigilance, and eye movements belong to WAF Battery.

**Figure 4 brainsci-12-00265-f004:**
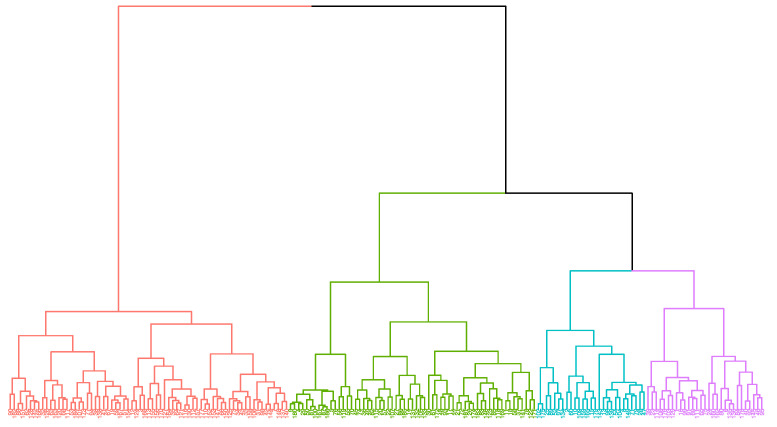
Dendrogram from the classification with Ward’s linkage algorithm (four clusters).

**Figure 5 brainsci-12-00265-f005:**
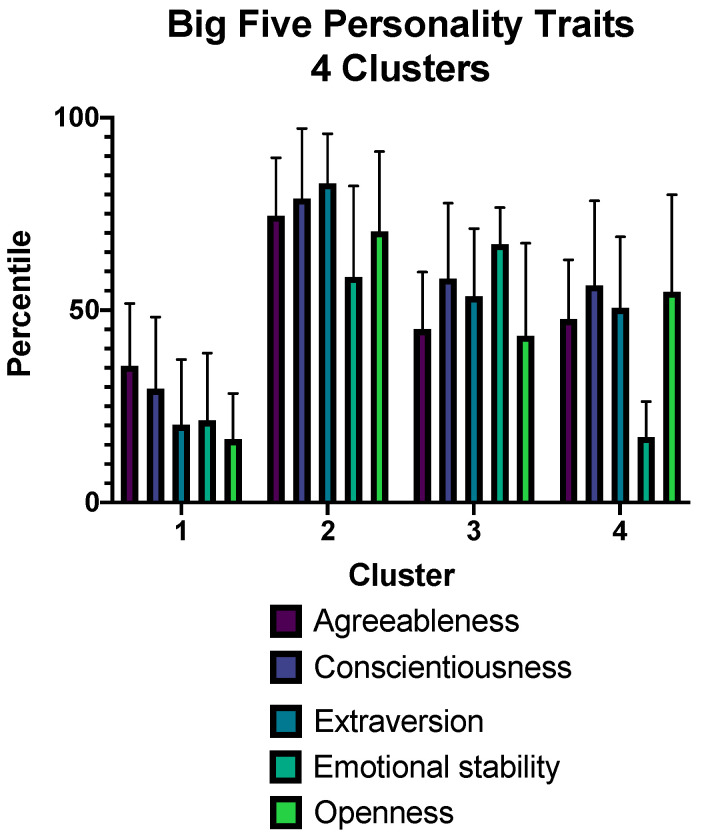
Distribution of Big Five personality traits between four clusters.

**Table 1 brainsci-12-00265-t001:** Main clinical and demographic characteristics of our sample of patients with post-COVID syndrome.

	Post-COVID Syndrome
Age	50.39 ± 11.26
Sex (women)	66 (71.0%)
Years of schooling	14.38 ± 3.64
BDI-II	14.03 ± 8.48
STAI-S	40.86 ± 11.57
STAI-T	48.18 ± 10.52
MFIS	52.96 ± 15.14
PSQI	9.96 ± 4.74
BSIT	8.92 ± 2.50

BDI-II: Beck Depression Inventory-II; BSIT: Brief Smell Identification Test; MFIS: Modified Fatigue Impact Scale; PSQI: Pittsburgh Sleep Quality Index; STAI: State-Trait Anxiety Inventory; STAI-S: STAI state anxiety subscale; STAI-T: STAI trait anxiety subscale.

**Table 2 brainsci-12-00265-t002:** Comparison of Big Five personality traits between patients with post-COVID syndrome and controls.

	Post-COVID Syndrome	Healthy Controls	t (*p* Value)
Agreeableness	47.94 ± 22.47	55.66 ± 22.40	2.34 (0.020)
Conscientiousness	50.33 ± 29.05	58.38 ± 26.82	1.96 (0.051)
Extraversion	45.04 ± 30.79	56.38 ± 29.78	2.55 (0.012)
Emotional stability	34.15 ± 27.00	45.24 ± 27.00	2.79 **(0.006)**
Openness	40.02 ± 31.45	48.62 ± 28.09	1.96 (0.051)

Statistically significant *p* values after false discovery rate correction for multiple comparisons are shown in bold.

**Table 3 brainsci-12-00265-t003:** Comparison of Big Five personality traits subfactors between patients with post-COVID syndrome and controls.

		Post-COVID Syndrome	Healthy Controls	t (*p* Value)
Agreeableness	Willingness to trust	39.94 ± 27.26	48.10 ± 29.01	1.97 (0.050)
	Genuineness	54.58 ± 26.27	60.10 ± 26.52	1.42 (0.156)
	Helpfulness	65.03 ± 27.38	74.15 ± 23.25	2.44 (0.015)
	Obligingness	61.39 ± 28.49	67.72 ± 23.97	1.64 (0.103)
	Modesty	33.34 ± 22.02	41.54 ± 23.39	2.45 (0.015)
	Good-naturedness	57.10 ± 27.88	65.14 ± 26.19	2.02 (0.044)
Conscientiousness	Competence	46.17 ± 26.45	55.60 ± 25.27	2.48 (0.014)
	Love of order	31.43 ± 26.12	41.11 ± 30.96	2.30 (0.022)
	Sense of duty	55.72 ± 24.87	61.59 ± 23.84	1.64 (0.102)
	Ambition	53.65 ± 26.94	59.26 ± 24.36	1.49 (0.138)
	Discipline	50.71 ± 29.46	58.40 ± 25.90	1.80 (0.060)
	Caution	74.33 ± 22.96	76.53 ± 22.54	0.65 (0.512)
Extraversion	Friendliness	55.56 ± 25.94	64.11 ± 25.42	2.26 (0.024)
	Sociableness	60.52 ± 28.97	71.48 ± 26.25	2.70 (0.007)
	Assertiveness	38.22 ± 25.25	43.11 ± 25.57	1.31 (0.191)
	Dynamism	57.48 ± 32.21	66.08 ± 28.84	1.91 (0.057)
	Adventurousness	24.29 ± 24.71	26.74 ± 23.12	0.69 (0.486)
	Cheerfulness	45.69 ± 34.00	58.38 ± 32.65	2.59 (0.010)
Emotional stability	Careefreeness	26.29 ± 25.42	29.43 ± 25.79	0.83 (0.404)
	Equanimity	28.61 ± 22.21	39.13 ± 24.05	3.09 **(0.002)**
	Positive mood	32.70 ± 29.71	45.60 ± 29.52	2.97 **(0.003)**
	Social confidence	37.52 ± 27.06	46.18 ± 28.28	2.13 (0.034)
	Self-control	46.81 ± 23.67	57.09 ± 25.50	2.84 **(0.005)**
	Emotional robustness	26.85 ± 25.63	35.05 ± 26.30	2.15 (0.033)
Openness	Openness to imagination	45.83 ± 32.17	54.41 ± 28.24	1.93 (0.055)
	Openness to aesthetics	25.80 ± 20.72	27.87 ± 21.18	0.675 (0.500)
	Openness to feelings	62.63 ± 31.14	65.43 ± 29.38	0.630 (0.530)
	Openness to actions	35.13 ± 31.07	44.54 ± 29.18	2.12 (0.035)
	Openness to ideas	54.33 ± 34.68	65.73 ± 29.79	2.40 (0.017)
	Openness to values	38.16 ± 24.32	44.88 ± 25.37	1.84 (0.067)

Statistically significant *p* values after false discovery rate correction for multiple comparisons are shown in bold.

**Table 4 brainsci-12-00265-t004:** Automatic linear modelling analysis assessing the personality predictors of depression and anxiety.

R^2^	Variables (Transformed)	Beta Coefficient	SE	t	95% CI	*p* Value	Importance
BDI-II (depression)							
0.363	Intercept	10.557	2.601	4.059	5.384, 15.731	<0.001	-
	Openness to ideas	−0.16	0.042	−3.86		<0.001	0.32
	Obligingness	0.089	0.031	2.881	0.027, 0.150	0.05	0.178
	Dynamism	−0.078	0.030	−2.627	−0.137, −0.019	0.010	0.148
	Openness to feelings	0.078	0.036	2.132	0.005, 0.150	0.036	0.098
STAI-S (anxiety)							
0.410	Intercept	38.042	3.496	10.881	31.083, 45.001	<0.001	-
	Openness to actions	−0.222	0.063	−3.515	−0.347, −0.096	0.001	0.153
	Caution	−0.209	0.060	−3.494	−0.328, −0.090	0.001	0.151
	Love of order	0.144	0.050	2.886	0.045, 0.243	0.005	0.103
	Competence	−0.232	0.082	−2.846	−0.395, −0.070	0.006	0.100
	Cheerfulness	−0.108	0.042	−2.568	−0.192, −0.024	0.012	0.082
	Social confidence	0.185	0.072	2.562	0.041, 0.328	0.012	0.081
	Adventurousness	0.196	0.078	2.497	0.040, 0.352	0.015	0.077
	Discipline	0.136	0.058	2.335	0.020, 0.252	0.022	0.068
	Assertiveness	−0.122	0.059	−2.073	−0.239, −0.005	0.041	0.053
	Openness to aesthetics	0.131	0.064	2.030	0.003, 0.259	0.046	0.051

Importance of each variable (predictor) is the residual sum of squares with the predictor removed from the model. Values were normalized in order for the sum of the predictors to be 1.

**Table 5 brainsci-12-00265-t005:** Neuropsychiatric characteristics of the four personality profiles derived from cluster analysis.

	Cluster 1	Cluster 2	Cluster 3	Cluster 4	F (*p* Value)
Number of patients with COVID-19/number of controls	43/25	23/39	13/14	14/15	8.96 * **(0.030)**
Age	53.74 ± 9.57	47.43 ± 10.88	52.83 ± 13.40	43.64 ± 11.46	4.02 **(0.010) ^c^**
Sex (women)	29 (67.4%)	17 (73.9%)	8 (61.5%)	12 (85.7%)	2.57 * (0.461)
BDI-II	16.72 ± 8.77	10.30 ± 6.69	9.67 ± 7.79	15.77 ± 7.60	4.66 **(0.005) ^a,b^**
STAI-S	24.51 ± 11.11	13.39 ± 8.78	19.00 ± 12.71	23.50 ± 10.71	5.74 **(0.001) ^a,d^**
STAI-T	32.65 ± 8.28	22.70 ± 9.33	19.58 ± 12.09	30.86 ± 9.32	9.88 **(<0.001) ^a,b,c,e^**
MFIS	55.07 ± 13.14	48.35 ± 17.29	50.67 ± 18.47	56.23 ± 13.42	1.29 (0.281)
PSQI	10.07 ± 4.93	9.82 ± 4.87	8.45 ± 4.52	11.00 ± 4.26	0.59 (0.617)
BSIT	8.98 ± 2.45	9.17 ± 2.05	8.75 ± 3.30	8.50 ± 2.79	0.229 (0.876)
SDMT	43.21 ± 15.31	41.17 ± 11.09	42.75 ± 11.45	47.29 ± 11.47	0.616 (0.606)
BNT	51.86 ± 5.61	50.65 ± 6.28	50.92 ± 7.63	53.07 ± 3.68	0.579 (0.630)
FGT-LT	53.74 ± 9.57	47.43 ± 10.88	51.77 ± 13.39	43.64 ± 11.46	0.338 (0.798)
ROCF-c	33.18± 3.20	32.13 ± 3.52	32.83 ± 2.40	32.07 ± 5.70	0.576 (0.632)
JLO	20.65 ± 7.19	19.78 ± 5.14	21.67 ± 5.29	19.57 ± 7.34	0.314 (0.815)
NBV	8.90 ± 13.66	7.93 ± 7.91	9.10 ± 11.90	9.25 ± 11.45	0.035 (0.991)
TMT-B	49.22 ± 36.56	44.22 ± 18.22	49.04 ± 26.27	37.31 ± 8.40	0.560 (0.643)
ToL	13.23 ± 3.77	13.25 ± 2.14	10.70 ± 2.00	13.16 ± 3.99	1.650 (0.185)

BDI: Beck Depression Inventory; BNT: Boston Naming Test; BSIT: Brief Smell Identification Test; FGT: Figural Memory Test (Learning total); JLO: Judgment Line Orientation; MFIS: Modified Fatigue Impact Scale; NBV: N-Back verbal test; PSQI: Pittsburgh Sleep Quality Index; ROCF-c: Rey-Osterrieth Complex Figure (copy accuracy); SDMT: Symbol Digit Modalities Test; TMT-B: Trail Making test part B; ToL: Tower of London. *: Chi-squared test. ANOVA with Tukey post-hoc analysis showed statistically significant differences between clusters 1 and 2 (^a^), clusters 1 and 3 (^b^), clusters 1 and 4 (^c^), clusters 2 and 4 (^d^), and clusters 3 and 4 (^e^). Bold indicates statistically significant results.

## Data Availability

The datasets generated and analysed are available from the corresponding author on reasonable request.

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
