# Peer review of "Examining Association of Personality Characteristics and Neuropsychiatric Symptoms in Post-COVID Syndrome"

_brainsci, 2022, doi:10.3390/brainsci12020265_

Round 1

Reviewer 1 Report

p.1 line 17: After saying “post-COVID syndrome”, the Authors could consider adding “as defined by the WHO” as the post-COVID syndrome is still a new term at the current time.

p.1 line 17: When saying “matched controls”, perhaps it would be more clear if the Authors say “demographically-matched controls” if that’s the case?

p.2 Table 1: Please consider adding scores for the control group and between group statistical differences to present how the two analysed groups were comparable.

p.3 line 109: A brief description of cognitive assessment would be beneficial. The Authors could name the tests used and refer to the previous publication for further details.

p.3 line 118: Please specify whether the correlations were performed on each of the two groups separately or all participants together e.g. in case there were no statistical differences on particular scores between the group?

p.4: Please add a paragraph on the between-group comparisons of olfactory results and cognitive testing results. A table displaying the data would be also very informative.

p.6. Please explain all abbreviations used in figures 2 and 3 in figure captions.

p.8 Table 4. The term “importance” should be replaced by the appropriate statistical term.

p.9 Figure 4. Seems to be cut (?)

p.10 line 201. Perhaps the Authors should reconsider saying “development” of depression and anxiety as there was no longitudinal data analysed. Instead, a more appropriate terms seems to be “presence” of depressive and/or anxiety symptoms.

p.10 line 207: Please replace “a first group” with “the first group” and so on for other groups.

p.10. When the Authors use the term “depression” and “anxiety” throughout the Discussion section, do they mean that the participants clasified as showing post-COVID syndrom also met the criteria to be diagnosed with major depression or anxiety? Or do the Authors mean that the participants with post-COVID syndrom present elevated levels of depressive and/or anxiety symptoms as compared to the matched control group? Please paraphrase throughout the Discussion section and add appropriate statistical data to the results section for depression including meeting the threshold for diagnosing depression and reference to the source recomending the threshold used in the study.

p.11 Limitations section: Please add discussion on the limitations related to not considering the severity of the disease (i.e., COVID-19), treatment used, hospitalization (e.g. length), the number of times participant was diagnosed with COVID-19 or time since the last COVID-19 diagnosis.

p.11 line 223: Please consider addting at the end of the sentence “as compared to people who were never diagnosed with COVID-10” or similar phrase for clarity.

General comment: Please consider dedicating a few more sentences throughout the article to olfactory function and cognitive function as those are important characteristics in this patient population and could be of interest to the Audience.

Author Response

Dear Reviewer,

Attached you will find the revised version of our manuscript.

We thank the reviewer for the very interesting observations and contributions to our article. We here address each of their commentaries.

Comments and Suggestions for Authors

p.1 line 17: After saying “post-COVID syndrome”, the Authors could consider adding “as defined by the WHO” as the post-COVID syndrome is still a new term at the current time.

RESPONSE: We have modified the text as suggested.

p.1 line 17: When saying “matched controls”, perhaps it would be more clear if the Authors say “demographically-matched controls” if that’s the case?

 RESPONSE: We have modified the text as suggested.

p.2 Table 1: Please consider adding scores for the control group and between group statistical differences to present how the two analysed groups were comparable.

 RESPONSE: Thanks for this suggestion. Unfortunately, the scales BDI-II, STAI, MFIS, PSQI, and BSIT were not administered to the control group. We have added the age, sex, and years of schooling of the control group in the text.

p.3 line 109: A brief description of cognitive assessment would be beneficial. The Authors could name the tests used and refer to the previous publication for further details.

 RESPONSE: We have specified the cognitive tests used, as suggested.

p.3 line 118: Please specify whether the correlations were performed on each of the two groups separately or all participants together e.g. in case there were no statistical differences on particular scores between the group?

 RESPONSE: Correlations were performed within the post-COVID syndrome group. We have specified this information in the Statistical analysis section as suggested.

p.4: Please add a paragraph on the between-group comparisons of olfactory results and cognitive testing results. A table displaying the data would be also very informative.

 RESPONSE: We have completed the Table 5 with olfactory and main cognitive testing results.

p.6. Please explain all abbreviations used in figures 2 and 3 in figure captions.

RESPONSE: We have added the explanation for all abbreviations as suggested.  

p.8 Table 4. The term “importance” should be replaced by the appropriate statistical term.

 RESPONSE: We appreciate this suggestion very much. We have defined the term “importance” at the foot of the table. “Importance” is defined as the residual sum of squares with the predictor removed from the model, normalized so that the importance values sum to 1. We maintained the term “importance”, because this is the output of the SPSS.

p.9 Figure 4. Seems to be cut (?)

 RESPONSE: We have amended the Figure. This was a problem when loading the figure into the template. We have amended it.

p.10 line 201. Perhaps the Authors should reconsider saying “development” of depression and anxiety as there was no longitudinal data analysed. Instead, a more appropriate terms seems to be “presence” of depressive and/or anxiety symptoms.

 RESPONSE: Many thanks for this suggestion. We have replaced “development” by “presence” as suggested.

p.10 line 207: Please replace “a first group” with “the first group” and so on for other groups.

 RESPONSE: We have amended the text as suggested.

p.10. When the Authors use the term “depression” and “anxiety” throughout the Discussion section, do they mean that the participants clasified as showing post-COVID syndrom also met the criteria to be diagnosed with major depression or anxiety? Or do the Authors mean that the participants with post-COVID syndrom present elevated levels of depressive and/or anxiety symptoms as compared to the matched control group? Please paraphrase throughout the Discussion section and add appropriate statistical data to the results section for depression including meeting the threshold for diagnosing depression and reference to the source recomending the threshold used in the study.

RESPONSE: We have added the prevalence of moderate-severe depression and clinically significant anxiety according to the cutoffs of the scales administered in the patients with post-COVID syndrome to Table 1 (We have also modified the values of the STAI according to the international version of the scale). However, the diagnosis of major depressive disorder requires a specific assessment. Thus, we have modified the Discussion section following the reviewer’ suggestion. We have replaced “depression” by “depressive symptoms”, as suggested.

p.11 Limitations section: Please add discussion on the limitations related to not considering the severity of the disease (i.e., COVID-19), treatment used, hospitalization (e.g. length), the number of times participant was diagnosed with COVID-19 or time since the last COVID-19 diagnosis.

RESPONSE: We have added these Limitations to the Discussion section.  

p.11 line 223: Please consider adding at the end of the sentence “as compared to people who were never diagnosed with COVID-10” or similar phrase for clarity.

RESPONSE: Many thanks for this interesting clarification. We have added to the sentence in the Conclusions section.

General comment: Please consider dedicating a few more sentences throughout the article to olfactory function and cognitive function as those are important characteristics in this patient population and could be of interest to the Audience.

We have completed the Discussion section following this suggestion. We have a specific article under review evaluating cognitive function, and for this reason, in this study we have not deepened further.

Reviewer 2 Report

Thank you for the opportunity to review this article addressing the role of personality traits in patients with post-COVID syndrome. The article is relevant to a current topic, but the purpose of the study needs to be better explained and supported by literature on, for example, other infectious diseases or those affecting central nervous system functions (e.g., HIV)
The authors correctly note that personality traits are a relatively stable human disposition. Therefore, it is not clear why they believe that there is a correlation between post-Covid syndrome and measured personality dimensions. Did the authors expect a different pattern of correlation than in the control group that did not undergo Covid?  If so, what would be the mechanism explaining these differences?
It is also unclear why the correlation analysis also involved objective measures of neuropsychological impairment, such as olfactory impairment? This part of the analysis seems quite random and unsupported by previous studies. Correlation analysis of selected variables on a control group is also lacking. 
In discussion, the authors state that their results suggest that personality traits at least partially explain the development of depressive anxiety symptoms in the context of post-COVID-19 syndrome. One wonders if these results would not be better explained by the severity of the illness. Do the authors have such data and could they include them in their analyses?
Some wording seems inaccurate e.g. "reduced levels of the main personality traits" 

Author Response

Dear Reviewer,

Attached you will find the revised version of our manuscript.

We thank the reviewer for the very interesting observations and contributions to our article. We here address each of their commentaries.

Thank you for the opportunity to review this article addressing the role of personality traits in patients with post-COVID syndrome. The article is relevant to a current topic, but the purpose of the study needs to be better explained and supported by literature on, for example, other infectious diseases or those affecting central nervous system functions (e.g., HIV).

RESPONSE: We appreciate this interesting suggestion. We have completed the Introduction accordingly.

The authors correctly note that personality traits are a relatively stable human disposition. Therefore, it is not clear why they believe that there is a correlation between post-Covid syndrome and measured personality dimensions. Did the authors expect a different pattern of correlation than in the control group that did not undergo Covid?  If so, what would be the mechanism explaining these differences?

RESPONSE: Thanks for this comment. Post-COVID syndrome is a very heterogeneous disorder with many different symptoms. We evaluated the correlation between personality traits and these symptoms, aiming to know what symptoms are at least partially associated with personality. Interestingly, personality traits were associated with depression and anxiety, but not with cognitive dysfunction or fatigue. From our perspective, this is important for several reasons. First, personality could partially explain the presence of depression or anxiety in some patients. Second, cognitive dysfunction or fatigue do not depend on personality, so other mechanisms should be investigated. We do not try to demonstrate a different behavior between controls and post-COVID patients. In fact, the influence of personality in anxiety and depression has been shown in other chronic disorders and our findings are consistent with the literature in other disorders. We have modified some parts of the Introduction and Discussion following the reviewers’ suggestions.

It is also unclear why the correlation analysis also involved objective measures of neuropsychological impairment, such as olfactory impairment? This part of the analysis seems quite random and unsupported by previous studies. Correlation analysis of selected variables on a control group is also lacking. 

RESPONSE: We agree with the reviewer that olfactory impairment should have no relationship with personality. However, olfactory dysfunction is a frequent symptom in post-COVID patients, and we believe that, as our aim was to evaluate the association between personality and post-COVID syndrome symptoms, it makes sense to include this correlation. In addition, we believe that the absence of correlation with olfactory supports the other findings (absence of correlation with fatigue or cognitive function, but significant correlation with anxiety and depression).

Healthy subjects were not assessed with the scales of fatigue or olfactory testing, and depression or neuropsychiatric symptoms were not allowed Thus, we cannot provide the correlation analysis for comparison with post-COVID patients.

In discussion, the authors state that their results suggest that personality traits at least partially explain the development of depressive anxiety symptoms in the context of post-COVID-19 syndrome. One wonders if these results would not be better explained by the severity of the illness. Do the authors have such data and could they include them in their analyses?

RESPONSE: Many thanks for this interesting suggestion. When we started the study, there were no published scales evaluating the severity of post-covid syndrome. Thus, we have the data of severity that we have included in the study for the individual symptoms (fatigue, cognitive function, etc.). We have added this interesting suggestion as a limitation of the study.

Some wording seems inaccurate e.g. "reduced levels of the main personality traits" 

RESPONSE: Thanks for this comment. We have replaced by “lower scores”.

Reviewer 3 Report

Very interesting and novel manuscript on personality traits in post-COVID syndrome are associated with anxiety and depression, but not cognitive dysfunction or fatigue. well written and clear.  I could not open the supplementary file though.

  1. The title needs to be shorter please and Im not sure that "personality traits" is the right term. Please can you reconsider if there is a more apt set of words here as the words "personality traits" has a lot of stigma.
  2. Overall sound methodologically but in the Discussion there is a negative bias which indicates that the anxiety and depression etc associated with COVID has a negative effect on a persons personality. I thought personality was a "relatively fixed" phenomenon? Please reconsider the wording.
  3.  Please include some more refs from 2021 and 2022 as a lot has been written on the sequalae of COVID19 in the last 6 months
  4. Please run the manuscript through TurnItIn or another plagiarism checker as some of the phrasing in the Introduction seems familiar.

Author Response

Dear Reviewer,

Attached you will find the revised version of our manuscript.

We thank the reviewer for the very interesting observations and contributions to our article. We here address each of their commentaries.

Very interesting and novel manuscript on personality traits in post-COVID syndrome are associated with anxiety and depression, but not cognitive dysfunction or fatigue. well written and clear.  I could not open the supplementary file though.

  1. The title needs to be shorter please and Im not sure that "personality traits" is the right term. Please can you reconsider if there is a more apt set of words here as the words "personality traits" has a lot of stigma.

RESPONSE: Thanks for this suggestion. We have re-written the title. We now proposed the following title: “Personality assessment in post-COVID syndrome” to avoid potential stigma. “Personality traits” is a very common term in the literature, but we agree with the reviewer’s comment.

  1. Overall sound methodologically but in the Discussion there is a negative bias which indicates that the anxiety and depression etc associated with COVID has a negative effect on a persons personality. I thought personality was a "relatively fixed" phenomenon? Please reconsider the wording.

RESPONSE: Thanks for this comment. We agree. We have modified the text, specifying that we meant the responses to the questionnaire.

  1.  Please include some more refs from 2021 and 2022 as a lot has been written on the sequalae of COVID19 in the last 6 months

RESPONSE: We have completed the Reference list, adding several relevant and interesting papers published in the last few months, as suggested.

  1. Please run the manuscript through TurnItIn or another plagiarism checker as some of the phrasing in the Introduction seems familiar.

RESPONSE: Thanks for this suggestion. We have used a plagiarism checker, but less than 10% was detected, which is considered adequate.